# Comprehensive Analysis of Feature Extraction Methods for Emotion Recognition from Multichannel EEG Recordings

**DOI:** 10.3390/s23020915

**Published:** 2023-01-12

**Authors:** Rajamanickam Yuvaraj, Prasanth Thagavel, John Thomas, Jack Fogarty, Farhan Ali

**Affiliations:** 1National Institute of Education, Nanyang Technological University, Singapore 637616, Singapore; 2Interdisciplinary Graduate School, Nanyang Technological University, Singapore 639798, Singapore; 3Montreal Neurological Institute, McGill University, Montreal, QC H3A 2B4, Canada

**Keywords:** EEG, emotion recognition, EEG feature extraction, valence, arousal, pattern recognition

## Abstract

Advances in signal processing and machine learning have expedited electroencephalogram (EEG)-based emotion recognition research, and numerous EEG signal features have been investigated to detect or characterize human emotions. However, most studies in this area have used relatively small monocentric data and focused on a limited range of EEG features, making it difficult to compare the utility of different sets of EEG features for emotion recognition. This study addressed that by comparing the classification accuracy (performance) of a comprehensive range of EEG feature sets for identifying emotional states, in terms of valence and arousal. The classification accuracy of five EEG feature sets were investigated, including statistical features, fractal dimension (FD), Hjorth parameters, higher order spectra (HOS), and those derived using wavelet analysis. Performance was evaluated using two classifier methods, support vector machine (SVM) and classification and regression tree (CART), across five independent and publicly available datasets linking EEG to emotional states: MAHNOB-HCI, DEAP, SEED, AMIGOS, and DREAMER. The FD-CART feature-classification method attained the best mean classification accuracy for valence (85.06%) and arousal (84.55%) across the five datasets. The stability of these findings across the five different datasets also indicate that FD features derived from EEG data are reliable for emotion recognition. The results may lead to the possible development of an online feature extraction framework, thereby enabling the development of an EEG-based emotion recognition system in real time.

## 1. Introduction

Emotions have a complex and fundamental role in cognition and behavior, influencing how we interact with and interpret our daily life experiences. Technology that can help recognize and measure emotions is highly desirable, as this can facilitate research and development in areas such as healthcare, education, psychology, robotics, marketing, and entertainment. Emotion recognition technology can also offer individuals (or clinicians) tools to aid emotion regulation and intervention. However, despite years of interest in psychology and affective computing, the development of reliable and generalizable emotion detection techniques is still a challenge. To that end, this study provides a comprehensive analysis of electroencephalogram (EEG) measures of emotional states, categorized in terms of valence (positive vs. negative) and arousal (high vs. low).

Numerous experiments on emotion recognition have been undertaken in recent years utilizing both physiological signals (e.g., electrocardiogram (ECG), galvanic skin resistance (GSR), electromyogram (EMG), respiration rate (RR), electrodermal activity (EDA) and EEG signals) [1,2] and behavioral data (e.g., facial expression images, body gestures, speech and voice signals) [3,4]. Behavioral data can provide useful measures of emotion-related processes; however, they can also be easily biased due to their subjective and controllable nature. In comparison, physiological signals are relatively automatic and uncontrolled and, therefore, may capture processes that can distinguish an individual’s true (unbiased) emotional states more objectively. Thus, relative to behavioral data, physiological signal-based emotion recognition has fundamental advantages in terms of reliability and validity.

A range of physiological signals have been explored for emotion recognition [1,5]. However, relative to other modalities, techniques based on EEG data have received remarkable attention due to the direct link between EEG and the neurophysiological activity of the central nervous system, as well as its high time resolution and reliability. Furthermore, due to the rapid advancement of sensor technology EEG data collection is becoming more practical. Considering the popularity of EEG as a measure of emotion, and its increasing accessibility for researchers and consumers, this current study is focused on EEG-based emotion recognition. Despite the large number of studies conducted on EEG-based emotion recognition, there are unsolved issues and questions. For example, the number of emotion classes recognized, the number of electrodes used, the accuracy of emotion recognition, and the generalization of the emotion recognition task.

To detect emotions using EEG, researchers traditionally extract a range of signal properties referred to as ‘features’, which are then analyzed relative to emotions (or emotion processing), to explore their utility for detecting or classifying experienced emotions. The accuracy of emotion recognition will be largely influenced by the quality of feature extraction and their functional relevance (or significance) to emotions. Until now, few EEG studies have been performed to compare the importance of different EEG features that are often used for emotion recognition. Schaaff and Schultz [6] compared classification accuracy of pleasant, neutral, and unpleasant emotional state among two different sets of features measured in the time domain (e.g., statistical), and frequency domain (e.g., Fast Fourier Transform) . Petrantonakis et al. [7] computed the higher order crossing (HOC) features of EEG signals and evaluated their performance in classifying emotional states (such as happiness, surprise, anger, fear, disgust, and sadness) among statistical and wavelet-based features. Frantzidis et al. [8] suggested using the Event Related Potential (ERP) amplitude, ERP latency and Event Related Oscillation (ERO) amplitude as features for emotional state classification . In [9], the authors performed time–frequency analysis to assess the event related synchronization (ERS)/desynchronization (ERD) characteristics of EEG data and compared liking and disliking emotional states across various time–frequency ERS/ERD features. Jenke et al. [10] explored multiple features such as band power, HOC, fractal dimension (FD), discrete wavelet transform (DWT), Hilbert–Huang spectrum, differential asymmetry, and rational asymmetry. They compared the classification of happy, curious, angry, sad, and quiet emotional state among different feature vectors. Liu and Sourina [11] computed FD features and compared the performance with statistical features for valence recognition. Yuvaraj et al. [12] employed bispectrum features for basic emotional state classification and compared the classification performance with power spectrum, wavelet packet and nonlinear features. Recently, Nawaz et al. [13] proposed an emotion recognition framework based on the statistical features and evaluated the classification accuracy in comparison to power, wavelet, and entropy features. Together, these studies indicate the potential to detect or characterize basic emotional state using various EEG feature sets. However, it is difficult to compare the performance of feature sets across the different studies as most analyses were performed using only on handful of features, thus failing to provide insight into their relative utility for developing automated (i.e., online) systems that can understand or classify human emotions in applied settings. Furthermore, many studies were evaluated only on a monocentric data (i.e., single, or smaller dataset), typically collected from on a smaller cohort.

The present study aims to provide the most comprehensive analysis of different EEG feature sets for emotion recognition to date to determine which features are best for distinguishing emotional states, categorized in terms of valence and arousal. To achieve this, a wide range of features are analyzed across five public datasets to identify the most significant and generalizable EEG features distinguishing high/low emotional valence and arousal states; the five public datasets used in this study are, DEAP [14], DREAMER [15], MAHNOB-HCI [16], AMIGOS [17] and SEED [18]. The feature sets that are explored include statistical, fractal dimension (FD), Hjorth parameters, higher order spectra (HOS), and wavelet transform. Like in most machine learning studies (e.g., [7,12,13]), classification accuracy serves as the main performance metric in this investigation; however, given that machine learning accuracy can vary across classification techniques [7,12,13], we also test the performance of two common classifiers, specifically, support vector machine (SVM) and classification and regression tree (CART). In this way, we aim to recommend the most useful and generalizable EEG feature-classification technique for detecting emotional states and to guide the future development of emotion recognition systems.

The key contributions of the current study are the following: We (i) evaluated the performance using five independent and public EEG emotion datasets and (ii) identified the optimal feature set for reliable EEG-based emotional state recognition. To our knowledge, this study could be one of the first to utilize five independent public datasets to identify the optimal EEG feature set that can discriminate emotions. The rest of the work is arranged as follows. The scalp EEG datasets and the details of the method are explained in Section 2. Various experimental results and discussions are described in Section 3. Last, Section 4 covers the conclusions.

## 2. Materials and Methods

Figure 1 shows the methodological framework of the EEG and machine learning techniques used in the present study. For each EEG dataset (described in Section 2.1), the raw EEG data was subjected to (1) preprocessing, (2) feature extraction, and (3) emotional state classification based on ground-truth self-report data reflecting emotional valence and/or arousal.

### 2.1. Emotion-Related EEG Datasets

This study utilizes emotion-related EEG signals from the five most popularly used public datasets, namely MAHNOB-HCI, DEAP (Dataset for Emotion Analysis using Physiological signals), SEED (SJUT emotion EEG Dataset), AMIGOS (A dataset for Mood, personality, and affect research on individuals and GrOupS), and DREAMER. Table 1 summarizes the core details of these datasets that are relevant to the present research, including sample characteristics and EEG parameters. Datasets include EEG recordings from 15–40 (*M* = 27.4, *SD* = 9.4) young adult participants (55% male overall), recorded using different EEG systems with 14–62 scalp channels. The specifics of each dataset are described in detail in the subsequent paragraphs.

The MAHNOB-HCI was pioneered by Soleymani and fellows [16], which comprises of 32-channel EEG recordings and other peripheral nervous system (PNS) signals. The signals were obtained from 27 participants as they watched 20 video clips, which lasted from 34.9 s to 117 s. Participants rated their levels in valence, arousal, dominance, and predictability, after they watched each clip. The DEAP emotion dataset is a multimodal dataset created by [14], which comprises EEG signals from 32-channels and other PNS signals. These signals were collected from 32 healthy subjects when they were watching 40 music video clips (i.e., 40 trials in total), each video clip lasting a minute. After each video/trial, the participants were asked to rate their arousal, valence, dominance, like/dislike, and familiarity level using self-assessment report. The data of each video consists of 60-s EEG recordings and a 3 s baseline data. The EEG were collected with the sampling frequency *(Fs)* of 512 Hz. The SEED dataset [18] comprises EEG and eye movement signals from 15 participants exhibiting three different emotions namely positive, negative, and neutral emotional state. Each participant had three experiment sessions on different days. In each session, there were fifteen four-minute videos to evoke the required emotions. Therefore, for the three sessions, there are 45 trials in the database. The same fifteen videos were used in all three experiment sessions. The EEG signals were collected from 62 channels with *Fs* = 1000 Hz and down sampled to 200 Hz. After each session, the participants were asked to label the video according to the contents: −1 for negative, 0 for neutral, and 1 for positive. In this study, we employed only recordings with positive and negative labels from participants to assess our results with additional emotion datasets that apply binary classifiers.

The AMIGOS dataset [17] includes 14 channels of EEG data, 2 channels of ECG data, galvanic skin response, and frontal video. The dataset was prepared from the recordings of 40 participants when they viewed 16 film clips, which lasted no longer than 250 s. After seeing each movie clip, participants self-assessed their levels of arousal, valence, dominance, liking, familiarity, and seven fundamental emotions (happiness, disgust, surprise, fear, anger, sorrow and neutral). As stated in [19], seven participants (participant ID: 9, 12, 21, 22, 23, 24, 29, and 33) physiological signals had missing data, therefore we excluded them in our study. Some participants (participant ID: 5, 11, 28, and 30) did not have either valence or arousal affective state values, so we excluded their data as well. The DREAMER dataset was developed by [15] and comprised of EEG signals from 14 channels and 2-channel ECG signals. These signals were collected from 23 healthy participants (aged between 22 and 33 years) as they watched 18 video clips with lengths between 65 and 393 s. After every video clip, the participants assessed their degrees of arousal, valence, and dominance using self-assessment manikin (SAM). In addition, 60-second baseline signals were recorded before each clip. EEG signals were captured with an Emotive EPOC wireless neuro headset with *Fs* of 128 Hz.

In this study, only the raw EEG and self-report data reflecting emotional valence and arousal were extracted for analyses. Furthermore, to be consistent across datasets, only data from the first session was used from sets including multiple sessions (i.e., AMIGOS and SEED). Across all datasets, and in this study, emotional valence and arousal (Figure 2) were analyzed as two orthogonal dimensions [19,20,21], consistent with popular circumplex models of emotion (e.g., [22]). The self-report scales used to rate valence and arousal differed across datasets; for DEAP, MANHOB-HCI, and AMIGOS each dimension was rated on a scale of 1 to 9, whereas for DREAMER, they were rated from 1 to 5, with lower numbers reflecting more negative or lower valence and arousal, respectively. To test and validate EEG classification of emotion, EEG data were first categorized as either low or high valence and arousal relative to the midpoint of the respective self-report scale (e.g., DREAMER data with valence score <2.5 were classed as low valence and ≥2.5 were classed as high valence). For the SEED dataset, trials were already labeled in terms of positive (labeled 1) and negative (labeled 0) emotion categories; hence, further categorization was not necessary.

### 2.2. EEG Signal Preprocessing

EEG signal preprocessing, feature extraction, and emotional state classification were performed in Python (v3.7.1) and MATLAB (vR2020b). The average number of EEG trials across datasets was 569.6 (*SD* = 423.4), including 540 for MAHNOBHCI (20 trials × 27 participants), 1280 for DEAP (40 trials × 32 participants), 150 for SEED (10 trials × 15 participants), 464 for AMIGOS (16 trials × 28 participants), and 414 for DREAMER (18 trials × 23 participants). EEG trial data were filtered using a 50/60 Hz notch and 1 Hz high-pass Butterworth filters (4th order) to remove electrical mains and DC artefact. Data were then down sampled to 128 Hz to match the sample rates across datasets, before being rereferenced to the common average, and segmented into 2-second nonoverlapping epochs. Epochs were then subjected to automatic artefact rejection to remove eye-blinks and other electrical artefacts by excluding segments with data exceeding ±100 μV. There was an average of 1046 (*SD* = 411) epochs for valence and 1036 (*SD* = 438) epochs for arousal across participants that were accepted for further analysis.

### 2.3. EEG Feature Extraction

Feature extraction refers to the process of transforming raw data into numerical features that can be processed while preserving the information in the original data set. It yields better results than applying machine learning directly to the raw data. In the emotion recognition process through EEG signals, feature extraction is the crucial part of the emotion classification. The quality of the feature extraction will directly affect the accuracy of the emotion classification. In this study the feature extraction and analysis aimed to identify the salient EEG data that can distinguish or classify emotional states. To that end, we compared the classification performance of ten EEG feature sets that have shown reliable performance in previous emotion recognition studies [10,11,12,13,23], including Statistical, Wavelet, Fractal Dimension, Hjorth Parameters, Higher Order Spectra, Spectral Power, Entropy, Nonlinear, Connectivity, and Graph Metric features. For brevity, only the top five performing feature sets are reported in this article, including Statistical, Wavelet, Fractal Dimension, Hjorth Parameters, Higher Order Spectra features as described in Table 2. All feature sets were extracted from each channel and epoch of the preprocessed EEG data.

#### 2.3.1. Statistical Features

Descriptive statistical measures of EEG time-series data have been used for emotion recognition in previous studies [10,11,13]. In this study, the statistical feature set includes the mean ($\mu_{X}$), median ($\bar{{}_{X}}$), standard deviation ($\sigma_{X}$), mean of absolute values of 1st difference ($\delta_{X}$), 2nd difference($\gamma_{X}$), normalized 1st difference (${\bar{\delta }}_{X}$), and normalized 2nd difference (${\bar{\gamma }}_{X}$) measured from the time-series data at each channel, across epochs; these features were calculated as indicated in Equations (1)–(7) below:(1)$$\mu_{X}=\frac{1}{T}\sum_{t=1}^{T}\left(X\left(t\right)\right),$$
(2)$$\bar{X}=med\left(X\left(t\right)\right),$$
(3)$$\sigma_{X}=\sqrt{\frac{1}{T}\sum_{t=1}^{T}{\left(X\left(t\right)-\mu_{X}\right)}^{2}},$$
(4)$$\delta_{X}=\frac{1}{T-1}\sum_{t=1}^{T-1}|(X\left(t+1\right)-X\left(t\right)|,$$
(5)$$\gamma_{X}=\frac{1}{T-2}\sum_{t=1}^{T-2}|(X\left(t+2\right)-X\left(t\right)|,$$
(6)$${\bar{\delta }}_{X}=\frac{\delta_{X}}{\sigma_{X}},$$
(7)$${\bar{\gamma }}_{X}=\frac{\gamma_{X}}{\sigma_{X}}$$
where *X*(*t*) denotes the time series EEG signal and *T* represents the total number of EEG samples. In addition, we also extracted skewness, and kurtosis features from the EEG data.

#### 2.3.2. Wavelet Analysis

Wavelet transform is a popular time–frequency (TF) decomposition technique that divides the EEG signal in several approximation and details levels of wavelet coefficients corresponding to various EEG frequency ranges, while conserving the time information of the signal. Previous studies have used wavelet analysis to measure the EEG TF distribution related to emotions [13,24,25,26]. Here, six-level continuous wavelet transform (*CWT*) was applied using the Morlet window function to obtain wavelet coefficients of EEG bands. This mother wavelet is chosen based on its near optimal time–frequency (TF) representation characteristics [27]. Besides, Morlet wavelet is widely used in EEG-based emotion recognition studies [28,29]. For sampling rate of 128 samples/sec, we obtained 18 scales and extracted *CWT* coefficients from first 12 scales as they have frequency >1.25 Hz. Each scale frequencies are: 61.115 Hz, 43.59 Hz, 31.09 Hz, 22.17 Hz, 15.81 Hz, 11.28 Hz, 8.04 Hz, 5.74 Hz, 4.09 Hz, 2.92 Hz, 2.08 Hz, and 1.48 Hz. The equation used to compute the *CWT* coefficients from one-dimensional (1D) EEG signal data is given in Equation (Equation 8):(8)$$CWT\left(a,b\right)=\int_{-\infty }^{\infty }X\left(t\right)\frac{1}{\sqrt{\left|a\right|}}\psi \left(\frac{t-b}{a}\right)dt$$
where *x*(*t*) denotes the time-series EEG signal in this work, ψ is the mother wavelet, and *a* is the scaling parameter, and *b* is the shifting parameter. Since the coefficients extracted from this frequency range are related to emotion [27,28,29], we computed average of the absolute values of the wavelet coefficients in each level scales as wavelet features, which is defined in Equation (Equation 9):(9)$$\mu_{\left(C_{K,\ell }\right)}=\frac{1}{\ell }\sum_{\ell =1}^{N}{\left|C_{K,\ell }\right|}^{2},$$
(10)$$\sigma_{\left(C_{K,\ell }\right)}=\sqrt{\frac{\sum {\left(C\left(K,\ell \right)-\mu_{\left(C_{K,\ell }\right)}\right)}^{2}}{N}}$$
where $C_{\left(k,l\right)}$ denotes the each value of the wavelet coefficients at the *k*th decomposition level, *ℓ* is the number of coefficients, and *k* = 1, 2, 3⋯, *N* represents the number of decomposition levels.

#### 2.3.3. Fractal Dimension

FD features approximate the complexity (or fractality) of the EEG times-series data providing an indication the level of self-similarity of the EEG signal across all time scales. Previously, FD features have shown promise for EEG-based emotion recognition [13,23,30,31]. In this study, we considered several FD algorithms commonly used for EEG signal analysis, namely Katz [32], Petrosian [33], and Higuchi [34]; these algorithms are explained below.

*Katz’s fractal dimension (KFD)*: Katz suggested an algorithm to compute FD based on waveform planar curve [35], which is defined in Equation (Equation 11) as:(11)$$KFD=\frac{log(L)}{log(d)}$$
where, *d* represents the distance between the two consecutive points (curve diameter) and *L* denotes the curve length. The mean of FD is calculated as KFD, by dividing *L* and *d* by the mean distance between the locations (*a*), as shown in Equation (Equation 12):(12)$$KFD=\frac{log(L)}{log(d)}=\frac{log\left(\frac{L}{a}\right)}{log\left(\frac{d}{a}\right)}=\frac{log(N)}{log\left(N\right)+log\left(\frac{d}{L}\right)}$$
where *N* is the number of time samples in the EEG epoch.

*Petrosian fractal dimension (PFD)*: This algorithm converts time-series EEG signal into binary sequences [35]. The PFD is calculated as shown in Equation (Equation 13):(13)$$PFD=\frac{log(m)}{log\left(m\right)+log\left(\frac{m}{m+0.4N_{\delta }}\right)}$$
where $N_{\delta }$ denotes the number of segment pairs in the binary sequence that are not identical, and *m* represents the samples number in the segment.

*Higuchi’s fractal dimension (HFD)*: Higuchi developed a method for finding FD directly from the original time series by decomposing into *N* samples, *X*(*n*) = *X*(1), *X*(2), *X*(3)⋯ *X*(*N*). A new time-series signal is generated by selecting one sample after every *i*th sample, which is defined as:(14)$$X_{i}^{j}=X\left(i\right),X\left(i+j\right),X\left(i+2j\right),\ldots X\left(i+int\left(\frac{N-i}{j}\right)*j\right)$$
where *i* = 1, 2, 3, 4⋯*j*. Here, *i* represents the initial time, *j* represents the internal time, and *N* represents the total number of samples. For each *i*, the length of the curve, $L_{i}$(*j*) is represented as Equation (Equation 15), and then taken as the average value of *j* values of $L_{i}$(*j*).
(15)$$L_{i}\left(j\right)=\frac{\sum_{m=1}^{int\left(\frac{N-i}{j}\right)}\left|X\left(i+mj\right)-X\left(i+m-1\right)j\right|*\left(n-1\right)}{k*int\left(\frac{N-i}{j}\right)}$$

The HFD method is developed from the concept that the curve under consideration is fractal-like if $L\left(j\right)\alpha j^{\left(-FD\right)}$ where FD denotes fractal dimension, and it is measured as given in Equation (Equation 16):(16)$$HFD=\frac{<L(j)>}{logj}$$

#### 2.3.4. Hjorth Parameters

Hjorth parameters are statistical functions that explain the EEG signal characteristics in the time domain, which have also been successfully used in emotion recognition from EEG signals [10,36]. It consists of two main measures, namely mobility ($h_{1}$), and complexity ($h_{2}$) features [37,38], which are defined according to the following Equations (Equation 17) and (Equation 18) :(17)$$Mobility\left(h_{1}\right)=\sqrt{\frac{\sigma_{d}^{2}}{\sigma_{x(t)}^{2}}}=\frac{\sigma_{d}}{\sigma_{x(t)}},$$
(18)$$Complexity\left(h_{2}\right)=\sqrt{\frac{\frac{\sigma_{dd}^{2}}{\sigma_{d}^{2}}}{\frac{\sigma_{d}^{2}}{\sigma_{x(t)}^{2}}}}=\frac{\frac{\sigma_{dd}}{\sigma_{d}}}{\frac{\sigma_{d}}{\sigma_{x(t)}}}$$
where, *x*(*t*) represents the time-series EEG signal with a length of *N*, $\sigma_{x(t)}$ relates to the standard deviation (SD) of EEG signal, $\sigma_{x(t)}^{2}$ denotes the variance in the time-series EEG signal, $\sigma_{d}$ denotes the SD of the 1st derivative of *x*(*t*), and $\sigma_{dd}$ denotes the SD of 2nd derivative of *x*(*t*). This activity is mobility (estimates the mean frequency), and complexity (computes the bandwidth of the signal).

#### 2.3.5. Higher Order Spectra

Higher order spectra (HOS) are a spectral representation of higher order statistics that can retain the information related to deviations from Gaussianity and the degree of nonlinearity in the time-series EEG signal. Among the group of HOS features, bispectrum (*Bis*) is regarded as an effective feature for recognizing emotion from EEG signals [10,12,24]. Bispectrum depicts the Fourier Transform (FT) of the third order moment of the signal [39], calculated as shown in Equation (Equation 19).
(19)$$Bis\left(f_{1},f_{2}\right)=E\left[X\left(f_{1}\right)\cdot X\left(f_{2}\right)\cdot X^{*}\left(f_{1}+f_{2}\right)\right]$$
where *X*(*f*) is the FT of the given signal *X*(*t*), * represents its complex conjugate, and E[·] denotes the expectation operation. In this study, bispectrum features namely, bispectrum mean magnitude $\left(Bis_{Mag}\right)$, and different bispectrum moments were extracted from EEG segments [40], which are computed as Equations (19)–(22):

Bispectrum magnitude, $Bis_{Mag}$
(20)$$Bis_{Mag}=\frac{1}{N}\sum_{\Omega }\left|Bis\left(f_{1},f_{2}\right)\right|,$$

Bispectrum logarithmic amplitudes summation, $H_{1}$
(21)$$H_{1}=\sum_{\Omega }log(|Bis\left(f_{1},f_{2}\right)\left|\right),$$

Bispectrum logarithmic amplitudes of diagonal elements summation, $H_{2}$
(22)$$H_{2}=\sum_{\Omega }log(|Bis\left(f_{D},f_{D}\right)\left|\right),$$

1st order spectral moment of amplitudes of diagonal elements of the bispectrum, $H_{3}$
(23)$$H_{3}=\sum_{\Omega }mlog(|Bis\left(f_{D},f_{D}\right)\left|\right)$$
where *N* is the total number of time points in the principal domain region, Ω.

### 2.4. Emotion and EEG Feature-Classification Techniques

Two classification techniques, SVM and CART, were applied and evaluated for emotional valence and arousal recognition using each EEG feature set described above, as well as a combination of all feature sets; the specific combination of a feature set (e.g., statistical) and classifier (e.g., SVM) is considered a unique feature-classification technique, which can be tested relative to other combinations. In terms of the classifiers, SVM forms a decision boundary between two classes (e.g., low vs. high valence) and attempts to increase each class distance from the decision boundary [12]. The function of kernel is to take data as input and transform it into the required form. Different SVM algorithms use different types of kernel functions. In the current study, Gaussian radial basis function (RBF) SVM (GSVM) is used due its excellent learning performance [41] in many applications including EEG-based emotion recognition [12,42,43]. CART classifiers use a minimum cost-complexity pruning technique [44]. For example, every test could consist of a linear combination of attribute values for numeric attributes. As a result, the output tree shows a hierarchy of linear models [44]. We compared the performance of four classifiers that have shown reliable classification performance in previous EEG-based emotion recognition studies [5,13], including CART, GSVM, Random forest (RF) and k-nearest neighbor (KNN). For brevity, only top two performing classifiers are reported in this paper, including CART and GSVM. We applied Bayesian optimization technique for GSVM and CART classifiers to optimize the hyperparameters for each inner fold. For GSVM, we optimized 2 hyperparameters, namely box constrain and kernel scale. For CART, we optimized number of learning cycles, and learn rate, and minimum leaf size. Besides, we have used also random under sampling boosting for ensemble to effectively handle imbalanced data, and standardized the predictor data.

### 2.5. EEG Feature-Classification Accuracy

The accuracy of each EEG feature set-classification technique was evaluated using 4-fold cross-validation. In this approach, each participant’s data was divided into 4-folds (i.e., four equal subsets of their data without overlap); 3-folds are randomly used for classifier training and the remaining fold is used as the final test for accuracy and validation. This 4-fold process is performed four times so that each fold is used as a test set, resulting in four classifier accuracy scores for each feature-classification method and participant. The mean accuracy is then computed across the 4-folds reflecting the final feature-classification accuracy per participant. This is applied separately for each dataset. To evaluate overall emotion feature-classification performance, the mean, and the SD of the final accuracy scores were computed across all participants.

### 2.6. Statistical Analysis: Comparing Feature-Classification Performance between EEG Feature Sets

Two-tailed paired-sample *t*-tests were used to evaluate whether emotion classification performance differs between feature sets. Cliff’s Delta value was also computed as an additional effect-size measure of the difference between sets. It is a non-parametric effect size measure that computes the degree of difference between two groups of data (in this case, FD versus each feature set) beyond the meaning of *p*-values. Cliff’s Delta range between −1 and 1, with effect sizes of −1 or 1 indicating that there is no overlap between the two groups, whereas a 0.0 indicates no difference between feature set means. Statistical significance was defined as *p*-value < 0.05. The *p*-values were corrected for multiple comparisons using Holm–Bonferroni correction.

### 2.7. EEG Scalp Topography Related to Emotion Processing

The topographic distribution of the most significant feature sets was visually inspected to consider the spatial distributions associated with high/low valence and arousal. To improve visual comparison, the features from each dataset were standardized (z-scored) and only common channels that were shared by all datasets (i.e., 14 channels) were plotted.

## 3. Experimental Results and Discussion

This section presents the classification accuracy of different feature sets and classifiers for each public EEG dataset. Higher accuracy scores are indicative of feature-classification methods that are more reliable for EEG emotion recognition. Table 3 and Table 4 display the mean classification accuracy for emotional valence and arousal, respectively. Accuracy scores are shown for each feature-classification technique, including the combination of all feature sets (i.e., Combined-ALL); the highest accuracy scores within and across each dataset (i.e., Average) are highlighted in bold.

The majority of EEG feature-classification methods performed reasonably well with average classification accuracies ≥77.78% and 77.59% for valence and arousal, respectively. This is interesting as it suggests a complex relationship between 2D emotional states and many properties of the EEG signal and is consistent with the successful application of these features across previous emotion recognition studies [7,10,13]. As demonstrated by the average classification accuracy across datasets in Table 3 and Table 4, the performance of EEG FD feature set was higher for classifying high/low emotional valence and arousal relative to other features when using either the GSVM or CART classifiers. These results are broadly consistent with previous research highlighting the value of FD features for detecting implicit emotional states [31,45,46]. Furthermore, the FD feature set delivered classification results with the lowest SD of accuracy, showing that they perform more consistently than other techniques in this study; this is a valuable property, suggesting greater stability or reliability of this feature set for applied emotion recognition. This outcome is also consistent with prior research showing that the intraclass correlation coefficient (ICC) of FD features is higher for emotional state classification relative to other methods, supporting its reliability for categorizing valence and arousal [46].

Another important finding of the present study is that CART classifiers performed better for EEG emotion recognition compared to GSVM. Using the FD feature set, we achieved the highest mean classification accuracy (average across the five datasets) for valence and arousal as 85.06% and 84.55%, with CART classifier (hereafter named as FD-CART). This was found to be the case across all datasets utilized in this study and is in line with previous research supporting the utility of CART for emotion recognition [47,48]. For that reason, we focus on reporting feature set outcomes utilizing the CART classifier in subsequent sections. Figure 3 shows the box plot of the top three feature set (fractal dimension, wavelet transform, and statistical features) accuracies using CART on each dataset. The plot visually displays the distribution of classification accuracies across each subject and illustrates that the reliability of selected EEG feature set for applied emotion recognition. Table 5 summarizes the statistical results of two-tailed paired *t*-tests comparing CART classification performance between different feature sets. The *t*-test outcomes (*p*-values) and Cliff’s Delta effect size also demonstrate that FD features have significantly higher accuracy than other feature sets, confirming the descriptive observations in Table 3 and Table 4.

Table 5 shows the statistical results of two-tailed paired *t*-test of CART classification performances among different feature sets. From the *p*-value, it is clear that the results from the FD are statistically different (*p*-value < 0.05) from other feature sets listed in the table, including the combination of features. Table 5 also provides the mean difference effect size for paired samples based on Cliff’s Delta. From the Cliff’s Delta, it is apparent that, across the five datasets, the emotional state classification accuracy with FD feature set is more accurate on average. However, it is not always the most accurate in each individual case.

The present research utilized a data-driven approach for identifying EEG features that are optimal for emotion detection, thus while FD clearly demonstrates the best performance, it is currently unclear why this feature set is the most effective. Further research is needed to investigate this matter; however, considering our results and the prior literature, we speculate that methodological (technical) and/or functional reasons could explain why FD features are most effective for emotion recognition. In terms of methodology, FD features are nonlinear complexity estimators and calculated over short time-periods, are robust to noise, and do not require any prior transformation of the time series [46,49]. This differs to other methods (e.g., wavelet, statistical) and is beneficial for emotion recognition. At a functional level, fractality indicates whether the EEG signal is synchronous or repetitive over different time scales (i.e., similar patterns occur over shorter and longer intervals), representing the nonlinear complexity of underlying brain activity [50]. As explained by Zappasodi et al. [50] complexity is considered to reflect efficient neuronal functioning, varying between randomness and constant periodicity; with the extremes related to disfunction and difficulty shifting between brain states. From this viewpoint, we can speculate that different emotional states are associated with unique levels of signal complexity, with high/low valence and arousal leading to important shifts in network complexity on a spectrum. This is consistent with the idea that emotions can drive mental (and neuronal) states associated with more or less lability and/or cognitive flexibility (e.g., [51]). FD may provide a relevant and effective means to model those functional differences, which are not captured in other EEG measures of 2D emotional states.

The topography of FD features (i.e., KFD, PFD, and HFD) associated with high or low valence are plotted in Figure 4. The grand mean (GM) head maps calculated across datasets for KFD suggest that higher (i.e., more positive) valence was associated with less complexity (fractality) at frontal electrode sites, particularly over the left hemisphere, relative to periods of low valence. This pattern is somewhat consistent with the GM topography of PFD, which suggests higher valence is related to lower complexity at frontal, temporal, and occipital electrode sites. GM HFD indicates a slightly different pattern, with higher valence linked to relatively higher complexity at the most frontal EEG channels, but lower complexity over left frontocentral regions. In general, these results suggest that states of higher valence are related to less EEG complexity over frontal regions. However, this is not always consistent within datasets, and given the limited sites these topographic findings should be considered tentatively.

The topography of FD features (i.e., KFD, PFD, and HFD) associated with high or low arousal are plotted in Figure 5. The GM headmaps for KFD suggest that higher arousal is related to lower complexity at frontal and temporal sites over the left hemisphere. GM PFD is shows a similar spatial distribution for high and low arousal, with lower complexity at left frontocentral sites and temporoparietal sites relative to other scalp regions, and this pattern is stronger in periods of high arousal. GM HFD shows the opposite pattern compared to PFD. These GM topographic distributions are somewhat consistent with those shown for valence, with higher arousal broadly associated with lower complexity over the left hemisphere. However, it is important to note that these topographic interpretations are based only on visual inspection with limited channels. It is also apparent that these GM spatial distributions of FD features are not completely consistent across all datasets. For that reason, these topographic results should only be used as a tentative guide for research interested in FD distribution relative to emotional states or the optimal location for electrodes to facilitate EEG emotion recognition. For more definitive outcomes future research involving more EEG channels is needed.

Table 6 provides a comparison to other studies in the literature that have utilized more than one dataset to validate their methods. As the AMIGOS and DREAMER emotion datasets were only lately released, there are only limited comparative studies and hence, for comparison, baseline evaluation work also included in Table 5. Siddharth et al. [52] utilized RGB topographic maps computed from power spectral density (PSD) features using bicubic interpolation and assessed binary classification (low/high) for valence and arousal emotion using DEAP, DREAMER, MANHOB, and AMIGOS . They achieved results 71.09–83.02% for valence and 72.58–80.42% for arousal emotion recognition. In another study, Li et al. [53] suggested an approach that generates spatial maps from EEG signals and combined graph regularized extreme learning machine (GRELM) with SVM for recognizing emotions. They obtained an accuracy of 62.005–88.00% for valence emotion on DEAP and SEED emotion datasets. In the recent study, Topic and Russo [19] demonstrated a hybrid deep learning approach using holographic and topographic feature maps for emotion recognition using EEG signals. In this approach, they introduced EEG-topography in which they utilized the spatial and spectral information and performed classification of valence and arousal on DEAP, DREAMER, AMIGOS and SEED datasets. They reported 76.61–88.45% and 77.72–90.54% for valence and arousal emotion recognition, respectively. The AMIGOS emotion dataset [17] authors achieved the classification accuracy of 57.60% for valence state and 59.20% for arousal state using power spectral density (PSD) EEG features. Similarly, the researchers of the DREAMER dataset [15] achieved emotion recognition accuracy of 62.49% and 62.17% for valence and arousal, respectively. From all these studies, we can see that the identified FD feature set performs better than comparable methods previously reported for both affective states consistently in all the five datasets. This demonstrates the effectiveness of fractal dimension features combined with CART classifier for emotion recognition using EEG signals.

## 4. Conclusions

In this work, we present a comparative analysis on different feature extraction methods using multichannel EEG recordings for the creation of a reliable emotional state recognition system. A comprehensive set of features (statistical, FD, Hjorth parameters, HOS, and wavelet transform features) were obtained from the EEG signals. We conducted a quantitative comparison of feature extraction techniques with two different classifiers, GSVM, and CART. The emotion EEG datasets namely, DEAP, DREAMER, MAHNOB, AMIGOS and SEED were used to assess the performance of the study. The findings revealed that FD feature set are the most sensitive feature metric in distinguishing emotions categorized in terms of high/low valence and arousal. The FD-CART feature-classification method tested in this study achieves an overall best mean accuracy of 86.79% and 84.55% for binary classification of valence, and arousal, respectively, using all features in the FD set. Our results suggest that the fractality of the EEG time-domain data has a substantial role and is more reliable for emotional state recognition. This might result in the creation of an effective online framework for extracting EEG features and the development of a real-time human computer interactive system for emotional state recognition.

The study comes with two limitations. Firstly, it would be interesting to explore deep learning classifiers as an alternative for CART and SVM. In recent years, convolutional layers of deep neural networks have been found successful in EEG-based classification of emotion [54,55]. It was not feasible to explore this approach here due to lack of data. However, integrating deep learning with the present research may be a fruitful direction for further work in EEG emotion recognition. Secondly, though subject-dependent cross-validation approach is carried out, building a truly subject-independent (e.g., leave-one-subject-out) system would be more reliable and scalable. In the future, we will extend this approach to subject independent cross-validation (e.g., leave-one-subject-out) with emotional state categorization in three-dimensional space, i.e., the valence-arousal-dominance emotional model. In addition, we also intend to investigate the FD-CART feature-classification method on the combined emotion EEG datasets for training, validation, and evaluation purposes.

## Figures and Tables

**Figure 1 sensors-23-00915-f001:**
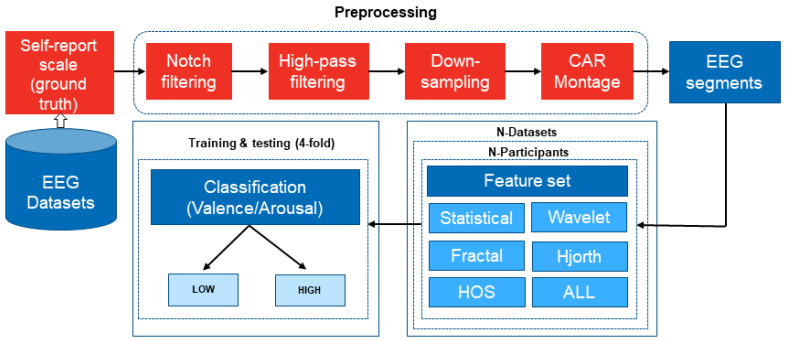
An overview of the proposed machine learning framework for emotion recognition based on EEG signals.

**Figure 2 sensors-23-00915-f002:**
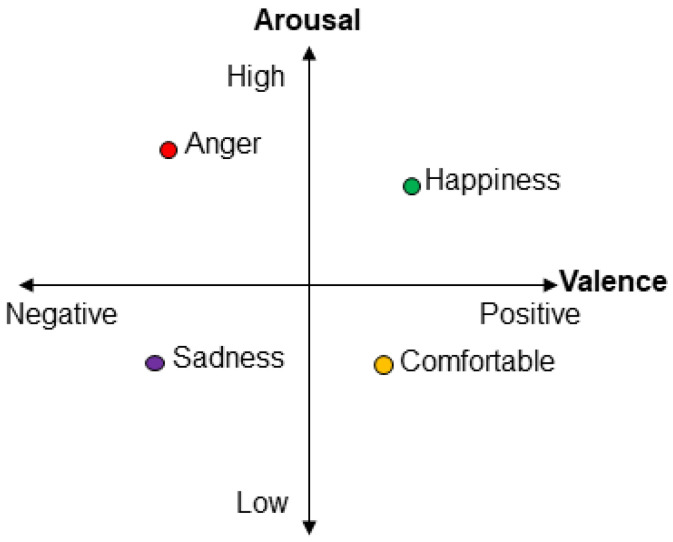
The two-dimensional model of emotions: valence–arousal plane.

**Figure 3 sensors-23-00915-f003:**
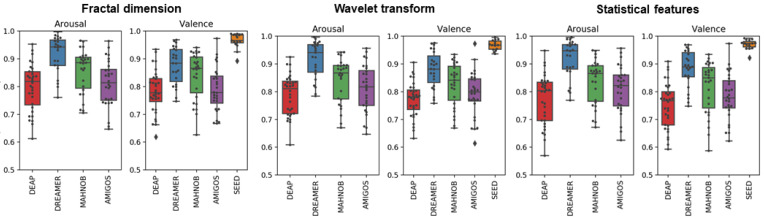
Top three feature sets. Boxplot of CART accuracy on each DEAP, DREAMER, MAHNOB, AMIGOS and SEED emotion dataset. *X*-axis represents the dataset name. *Y*-axis indicates the classification accuracy. Black dot in the figure represents average classification accuracy of each participant across 4-folds.

**Figure 4 sensors-23-00915-f004:**
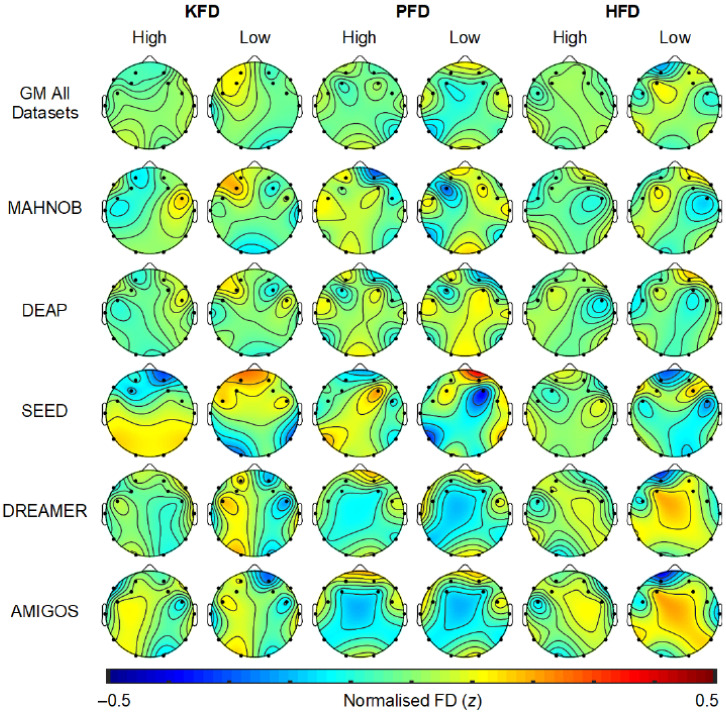
Topography of normalized EEG FD features for high/low valence. GM denotes the grand mean of each FD feature across all the datasets. KFD—Katz’s fractal dimension, PFD—Petrosian fractal dimension, HFD—Higuchi’s fractal dimension.

**Figure 5 sensors-23-00915-f005:**
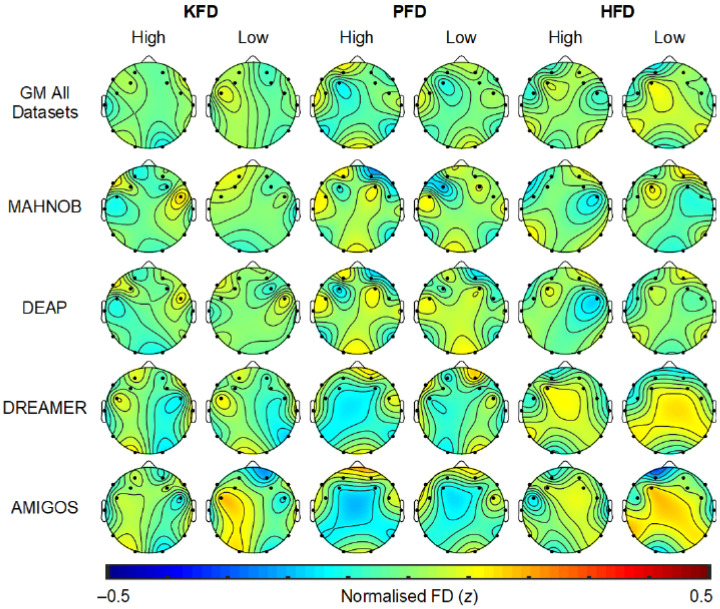
Topography of normalized EEG FD features for high/low arousal. SEED dataset does not have arousal class. GM denotes the grand mean of FD each feature across all the datasets.KFD-Katz’s fractal dimension, PFD-Petrosian fractal dimension, HFD-Higuchi’s fractal dimension.

**Table 1 sensors-23-00915-t001:** Information about the datasets used in this study.

Public Dataset Name	Pub. Year	Sample Size (N)	Gender Ratio (Mean Age ± SD)	Total Trials or Videos	Trial/ Video Dura.	Rec. Ses.	# EEG Channels /Device /Fs	Emotional States	Rating Scale Ranges (Thres.)
MAHNOB -HCI	2011	27	11M /16F (NS ± NS)	20	34.92 to 117 s	1	32/ BioSemi Active II /256 Hz	Valence & Arousal	1–9 (4.5)
DEAP	2012	32	16M/16F (26.9 ± NS)	40	60 s	1	32/ BioSemi Active II /512 Hz	Valence & Arousal	1–9 (4.5)
SEED	2015	15	7M/8F (23.27 ± 2.37)	10	∼240 s	3	62/ ESI Neuro Scan /1000 Hz	Positive & Negative	−1, 0, & 1 (NA)
AMIGOS	2018	40	27M/13F (28.3 ± NS)	16	<250 s	2	14/ Emotive EPOC /128 Hz	Valence & Arousal	1–9 (4.5)
DREAMER	2018	23	14M/9F (26.6 ± 2.7)	18	65–393 s	1	14/ Emotiv EPO /128 Hz	Valence & Arousal	1–5 (2.5)

All the EEGs are recorded using the international 10–20 positioning system. Fs = sampling frequency in Hz, M = Male, F = Female, SD = standard deviation, s = seconds, NS = not specified, NA = not applicable, Pub.—publication, dura.—duration, Rec.—recording, ses.—sessions, thres.—threshold.

**Table 2 sensors-23-00915-t002:** Summary of EEG features employed in this study.

Feature Set	Features	No. of Features
Statistical	Mean ($\mu_{x}$), Median ($\bar{{}_{X}}$), Standard deviation ($\sigma_{x}$), Skewness, Kurtosis, Mean of absolute values of 1st difference ($\delta_{x}$), Mean of absolute values of 2nd difference ($\gamma_{x}$), Normalized 1st difference (${\bar{\delta }}_{x}$), and Normalized 2nd difference (${\bar{\gamma }}_{x}$)	9
Wavelet	Mean and standard deviation of the absolute values of the coefficients in each of the 12 scales (with Morlet as mother wavelet).	24
Fractal dimension (FD)	Katz’s fractal dimension (KFD), Petrosian fractal dimension (PFD), and Higuchi’s fractal dimension (HFD).	3
Hjorth parameters	Mobility ($h_{1}$), and Complexity ($h_{2}$).	2
Higher order spectra (HOS)	Bispectrum magnitude ($Bis_{Mag}$), Sum of logarithmic amplitudes of Bispectrum ($H_{1}$), Sum of logarithmic amplitudes of diagonal elements in the bispectrum ($H_{2}$), and 1st-order spectral moment of amplitudes of diagonal elements of the bispectrum ($H_{3}$).	4

**Table 3 sensors-23-00915-t003:** Emotional Valence: Mean (±SD) EEG Feature-classification accuracy (%). Bold represents the highest average accuracy scores within and across each dataset.

Feature Set	Classifier	Dataset Name					Average
		DEAP	DREAMER	MAHNOB	AMIGOS	SEED	
Combined-ALL	GSVM	73.09 ± 0.060	86.56 ± 0.063	78.23 ± 0.080	76.94 ± 0.076	96.73 ± 0.024	82.31 ± 0.084
	CART	76.38 ± 0.072	**88.44 ± 0.066**	82.08 ± 0.081	78.47 ± 0.087	**97.08 ± 0.022**	84.49 ± 0.075
Statistical	GSVM	69.62 ± 0.066	83.74 ± 0.064	75.47 ± 0.077	74.47 ± 0.082	96.14 ± 0.035	79.89 ± 0.093
	CART	75.02 ± 0.086	88.26 ± 0.067	81.67 ± 0.097	78.19 ± 0.087	97.01 ± 0.020	84.03 ± 0.078
Wavelet	GSVM	69.11 ± 0.064	82.10 ± 0.075	71.94 ± 0.079	72.34 ± 0.070	93.39 ± 0.047	77.78 ± 0.089
	CART	77.34 ± 0.066	87.99 ± 0.066	82.57 ± 0.082	**79.12 ± 0.084**	96.78 ± 0.023	84.76 ± 0.070
Fractal dimension	GSVM	75.69 ± 0.065	83.94 ± 0.065	80.91 ± 0.078	76.83 ± 0.084	96.40 ± 0.030	82.75 ± 0.074
	CART	**78.18 ± 0.079**	87.59 ± 0.067	**83.98 ± 0.087**	79.07 ± 0.084	96.50 ± 0.030	**85.06 ± 0.066**
Hjorth parameters	GSVM	73.23 ± 0.068	82.20 ± 0.066	78.82 ± 0.069	71.57 ± 0.067	96.32 ± 0.023	80.43 ± 0.088
	CART	70.52 ± 0.063	80.86 ± 0.065	75.33 ± 0.071	70.21 ± 0.070	94.25 ± 0.037	78.23 ± 0.089
Higher order spectra	GSVM	73.78 ± 0.073	83.31 ± 0.076	79.39 ± 0.081	72.23 ± 0.086	96.83 ± 0.028	81.11 ± 0.088
	CART	72.18 ± 0.076	83.68 ± 0.069	78.27 ± 0.087	72.98 ± 0.082	95.66 ± 0.037	80.56 ± 0.086

**Table 4 sensors-23-00915-t004:** Emotional Arousal: Mean (±SD) EEG Feature-classification accuracy (%). Bold represents the highest average accuracy scores within and across each dataset.

Feature Set	Classifier	Dataset Name				Average
		DEAP	DREAMER	MAHNOB	AMIGOS	
Combined-ALL	GSVM	75.83 ± 0.072	90.35 ± 0.072	80.55 ± 0.081	79.49 ± 0.093	81.56 ± 0.053
	CART	78.82 ± 0.076	**92.02 ± 0.065**	83.21 ± 0.082	81.00 ± 0.092	83.76 ± 0.050
Statistical	GSVM	72.51 ± 0.085	88.92 ± 0.072	77.46 ± 0.084	77.10 ± 0.103	79.00 ± 0.060
	CART	77.38 ± 0.092	91.76 ± 0.068	83.72 ± 0.084	80.94 ± 0.087	83.45 ± 0.053
Wavelet	GSVM	71.60 ± 0.079	87.62 ± 0.089	75.65 ± 0.095	77.15 ± 0.096	78.01 ± 0.059
	CART	78.83 ± 0.077	91.60 ± 0.067	84.14 ± 0.082	**81.20 ± 0.087**	83.94 ± 0.048
Fractal dimension	GSVM	77.48 ± 0.072	88.99 ± 0.074	82.80 ± 0.079	79.10 ± 0.088	82.09 ± 0.044
	CART	**79.90 ± 0.086**	91.60 ± 0.067	**85.58 ± 0.085**	81.11 ± 0.087	**84.55 ± 0.045**
Hjorth parameters	GSVM	75.62 ± 0.069	87.02 ± 0.083	80.70 ± 0.075	75.28 ± 0.099	79.66 ± 0.047
	CART	73.14 ± 0.071	86.07 ± 0.085	76.94 ± 0.080	74.21 ± 0.107	77.59 ± 0.050
Higher order spectra	GSVM	75.83 ± 0.079	88.35 ± 0.082	81.13 ± 0.083	76.77 ± 0.091	80.52 ± 0.049
	CART	75.36 ± 0.081	88.69 ± 0.081	79.62 ± 0.086	76.79 ± 0.098	80.11 ± 0.051

N.B. The SEED dataset is not listed as it did not record arousal data.

**Table 5 sensors-23-00915-t005:** Statistical results (*p*-values effect size) of two-tailed paired *t*-test of CART classification performances among different feature sets.

Condition	*p*-Value		Cliff’s Delta Effect Size	
	Arousal	Valence	Arousal	Valence
FD vs. Wavelet	3.31 × 10^−3^	3.53 × 10^−2^	0.045	0.022
FD vs. Statistical	6.85 × 10^−8^	1.31 × 10^−9^	0.063	0.061
FD vs. Hjorth Parameters	2.13 × 10^−19^	1.22 × 10^−21^	0.397	0.420
FD vs. Higher order spectra	3.01 × 10^−19^	1.53 × 10^−21^	0.261	0.277
Combined-ALL	3.93 × 10^−4^	1.25 × 10^−3^	0.058	0.045

Effect size based on Cliff’s Delta. FD—Fractal dimension.

**Table 6 sensors-23-00915-t006:** Comparison with other studies in the literature.

Research Study	Features Employed Classification Method	Best Accuracy (%) Achieved				
		DEAP	DREAMER	MAHNOB	AMIGOS	SEED
Topic and Russo, [19]	HOLOfm CNN-SVM	V:76.61 A:77.72	V:88.20 A:90.43	-	V:87.39 A:90.54	V:88.45 A: -
Topic and Russo, [19]	TOPOfm CNN-SVM	V:76.30 A:76.54	V:81.96 A:84.92	-	V:80.63 A:85.75	V:70.37 A: -
Siddharth et al. [52]	RGB colored image CNN-ELM	V:71.09 A:72.58	V:78.99 A:79.23	V:80.77 A:80.42	V:83.02 A:79.13	-
Li et al. [53]	Spatial map GRELM-SVM	V:62.00 A: -	-	-	-	V:88.00 A: -
Katsigiannis et al. [15]	PSD SVM	-	V: 62.49 A:62.17	-	-	-
Miranda et al. [17]	PSD, SPA SVM	-	-	-	V:57.60 A:59.20	-
**This study**	**FD-CART**	**V:78.18** **A:79.90**	**V:87.59** **A:91.60**	**V:83.98** **A:85.58**	**V:79.07** **A:81.11**	**V:96.50** **A: -**

“-” means that this experiment was not in this research. A—Arousal, CNN—Convolutional Neural Network, ELM—Extreme Learning Machine; GELM—Graph regularized Extreme Learning Machine, HOLO-FM—Holographic Feature Maps, PSD—Power spectral Density, SPA—Spectral Power Asymmetry, SVM—Support Vector Machine, TOPO-FM—Topographic Feature Maps, V—Valence.

## Data Availability

Not applicable.

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
