# Peer review of "Comprehensive Analysis of Feature Extraction Methods for Emotion Recognition from Multichannel EEG Recordings"

_sensors, 2023, doi:10.3390/s23020915_

Round 1

Reviewer 1 Report

The manuscript is devoted to the solution of the problem of emotion recognition from EEG recording based on the joint use of both various feature extraction methods and classification techniques. The topic is interesting and actual. The paper is well-structured and contains all the necessary parts for this type of publication. To my mind, the manuscript can be accepted after minor revision. Below, I write my remarks.

1. The manuscript will look better, if at the end of the Introduction before the key contribution to allocate the unsolved parts of the general problem.

2. Subsection 2.3.2. Please, explain the reasonable using the Morlet wavelet in comparison with other types of wavelets. And, How did you optimize the level of wavelet decomposition? Please, add this information to the manuscript.

3. The conclusion section should be extended too by adding more concrete information regarding the obtained results with their analysis and further perspectives of their application.

Author Response

We, authors, express our sine gratitude to you for evaluating our article and providing suggestions for its improvement. We revised the article as per the reviewer comments and the revisions are provided here. 

Reviewer 2 Report

Why GSVM is considered? Explain in detail how GSVM and SVM differs.

There are several ML algorithms for classification such as Random forest, KNN, GBM, Why CART and GSVM is used? Justify

Explain in detail about  FD-CART. 

What is the difference between CART and Decision trees?

Comparison should be done with other tree based algorithms.

Explain in deatail about feature extraction and apllicationf of feature extraction.

What are the limitations of the study?

This is ensemble model of FD and CART. Compare with other ensemble algorithms.

Author Response

(The authors gave the same response as above.)
